# Temporal Self-Rewarding Language Models:
# Decoupling Chosen-Rejected via Past-Future

**Yidong Wang** [* 1] **Xin Wang** [* 1] **Cunxiang Wang** [* 2] **Junfeng Fang** [3] **Qiufeng Wang** [4] **Jianing Chu** [5]
**Xuran Meng** [6] **Shu-Xun Yang** [7] **Andrew Zhuoer Feng** [2] **Libo Qin** [8] **Wei Ye** [1] **Shikun Zhang** [1]

## Abstract

Self-Rewarding Language Models propose an architecture in which the Large Language Models(LLMs) both generates responses and evaluates its own outputs via LLM-as-a-Judge prompting, dynamically improving its generative capabilities through iterative Direct Preference Optimization (DPO). However, our analysis reveals a critical limitation in existing self-rewarding paradigms: the synchronized improvement of chosen and rejected responses progressively narrows the representational difference between contrasting samples, undermining effective preference learning. We propose **Temporal Self-Rewarding Language Models** that strategically coordinate past, present, and future model generations to sustain learning signals. Our dual-phase framework introduces: (1) *Anchored Rejection* - fixing rejected responses using the past initial model's outputs and (2) *Future-Guided Chosen* - dynamically curating chosen samples using next-generation model predictions. Extensive experiments across three model families (Llama, Qwen, Mistral) and different model sizes (Llama3B/8B/70B) demonstrate significant improvements when trained with our method compared to self-rewarding using same computation resources. For example, Llama3.1-8B reaches a 29.44 win rate on AlpacaEval 2.0 with our method, outperforming the self-rewarding baseline (19.69) by 9.75. Notably, our method also demonstrates superior out-of-distribution

generalization across mathematical reasoning (GSM8K), knowledge-based QA (ARC, TruthfulQA), and code generation (HumanEval) tasks, even though we do not specifically collect such training data. The generality of our temporal strategy is further validated by its benefits when extended to online reinforcement learning settings on mathematical reasoning tasks. Our code is publicly available at `https://github.com/TemporalSelfRewarding/TSR`.

## 1. Introduction

Large language models (LLMs) have attracted increasing attention in the field of artificial intelligence (OpenAI, 2023; Google, 2023; Zeng et al., 2022; Brown et al., 2020; Chowdhery et al., 2022; Anil et al., 2023; Wang et al., 2025; Li et al., 2025), with post-training (Stiennon et al., 2020; Ouyang et al., 2022; Bai et al., 2022)techniques proving particularly effective in enhancing model capabilities. Among various research methodologies, self-improvement(Huang et al., 2022; 2024; Yu et al., 2023; Qu et al., 2024; Wang et al., 2023) paradigms have emerged as a promising direction for autonomous model refinement. Recent advances in self-rewarding(Yuan et al., 2024) language models demonstrate an alternative paradigm to self-improvement, where language models serve dual roles as both response generators and evaluators (Yuan et al., 2024; Wu et al., 2024). Specifically, the self-rewarding paradigm builds upon the Supervised Fine-Tuned (SFT) model through an iterative optimization cycle that: (1) generating candidate responses to given prompts, (2) using the same LLM to evaluate these responses via LLM-as-a-Judge prompting (Zheng et al., 2023; Li et al., 2023a; Wang et al., 2024a), and (3) selecting preference pairs from the highest and lowest scoring responses for DPO training (Rafailov et al., 2023). Most existing work has focused on enhancing the model's judging capabilities to improve the effectiveness of the self-rewarding paradigm. For example, meta-rewarding approaches refine judgment skills through self-evaluation (Wu et al., 2024), while other methods include consistency regularization of reward models (Wang et al., 2024b), self-consistency mechanisms for

---

[*]Equal contribution [1]Peking University [2]Tsinghua University [3]National University of Singapore [4]Southeast University [5]North Carolina State University [6]University of Michigan [7]Beijing Institute of Technology [8]Harbin Institute of Technology, Shenzhen. Correspondence to: Cunxiang Wang <wangcunxiang303@gmail.com>, Wei Ye <wye@pku.edu.cn>, Shikun Zhang <zhangsk@pku.edu.cn>.

*Proceedings of the $43^{rd}$ International Conference on Machine Learning*, Seoul, South Korea. PMLR 306, 2026. Copyright 2026 by the author(s).

internal rewards (Zhou et al., 2025), and process-based evaluation for mathematical reasoning (Zhang et al., 2025). Unlike traditional approaches that rely on static reward models or fixed preference datasets, these methods allow for the continuous evolution of generation and evaluation quality.

Despite the success of self-rewarding language models on benchmarks like AlpacaEval (Li et al., 2023c) and Arena-Hard (Li et al., 2024), our theoretical analysis reveals a critical limitation: when the representational similarity between chosen and rejected responses increases, the DPO gradient vanishes, causing the training process to collapse. This theoretical prediction is empirically validated by our findings - as quantified in Fig. 1, the representations of chosen and rejected responses in the self-rewarding paradigm become progressively similar, with the score gap between them shrinking by 9 times during the same period (all responses evaluated by GPT-4o-mini to ensure consistent scoring and eliminate potential bias from varying judge capabilities across LLMs). This representational convergence directly leads to diminishing quality differences between generated answers, which in turn weakens or eliminates the learning signal for preference optimization. We attribute this convergence to reduced response diversity after reinforcement learning (Zhang et al., 2024; Kirk et al., 2023), which conflicts with the fundamental assumption of preference learning that requires clear quality differences between positive and negative samples for effective optimization (Lanchantin et al., 2025; Razin et al., 2025). The resulting vanishing gradient problem creates a vicious cycle where decreasing answer distinctness makes it harder to produce high-quality preference data, further exacerbating the learning signal deterioration.

To address the above issues, we propose **Temporal Self-Rewarding Language Models** that strategically coordinates past, present, and future model generations to maintain effective preference learning signals. Our approach consists of two key components: (1) *Anchored Rejection* that fixes rejected responses using outputs from the initial SFT model (past generation) to prevent quality inflation in negative samples, and (2) *Future-Guided Chosen* that selects high-quality positive samples by incorporating predictions from a future model version. The future model is obtained by first performing DPO training on the current model using the anchored rejection pairs, creating a temporary model that represents the next generation's capabilities. This future model then helps produce superior responses that would otherwise be unavailable to the current model. By decoupling the chosen and rejected responses through this temporal approach, our method maintains clear differences between good and bad examples during training, as shown in Figure 1. **To ensure fairness, we run half as many iterations as self-rewarding, which compensates for its doubled per-iteration cost due to training a future model, maintaining comparable total resource consumption.**

Extensive experimental results across three model families (Llama, Qwen, Mistral) and different model sizes(Llama3B/8B/70B) evaluated on multiple benchmarks (AlpacaEval 2.0, Arena-Hard-v0.1, MT-Bench) demonstrate the superior performance of our Temporal Self-Rewarding approach. On AlpacaEval 2.0, our method achieves a 29.44% win rate with Llama3.1-8B, outperforming the self-rewarding baseline (19.69%) by 9.75%. Similar improvements are observed on Arena-Hard-v0.1, where Qwen2.5-7B scores 34.4 with our method, exceeding the self-rewarding baseline (21.5) by 12.9%. The effectiveness of our temporal coordination strategy is further validated by the sustained quality gap between chosen and rejected responses throughout training iterations. Our method also demonstrates strong generalization across mathematical reasoning (GSM8K), knowledge-based QA (ARC, TruthfulQA), and code generation (HumanEval) tasks, with Temporal SR Iter1 achieving a 54.43% accuracy on TruthfulQA - 2.66% higher than the best self-rewarding. To further demonstrate the generality of our temporal coordination strategy, we extend our approach to online reinforcement learning and witness consistent performance improvements across mathematical reasoning benchmarks.

In conclusion, to the best of our knowledge, Temporal Self-Rewarding represents the first systematic approach to address the diminishing preference signal problem in self-rewarding language models through temporal coordination of model generations. Our method establishes a new paradigm for iterative self-improvement that maintains effective learning signals by strategically leveraging past, present, and future model capabilities. The proposed framework not only outperforms existing self-rewarding approaches but also provides insights into the dynamics of preference learning in iterative optimization settings. By decoupling the chosen and rejected samples, we enable more stable and effective model alignment while preserving the computational efficiency of the self-rewarding paradigm.

## 2. Methodology

In this section, we first present our theoretical analysis of the gradient collapse problem in self-rewarding models, then introduce our two-phase Temporal Self-Rewarding.

**Theoretical Analysis** We define a process where for a prompt $x$, a chosen response $\mathbf{y}_w$ and a rejected response $\mathbf{y}_l$ are generated from latent representations $\mathbf{h}_w \sim \pi_\theta^h(\cdot|x)$ and $\mathbf{h}_l \sim \pi_\theta^h(\cdot|x)$, respectively. Critically, the designation of these responses as "chosen" or "rejected" is performed by the same model, $\pi_\theta$, that generated them.

Our approach builds upon DPO, which directly optimizes a

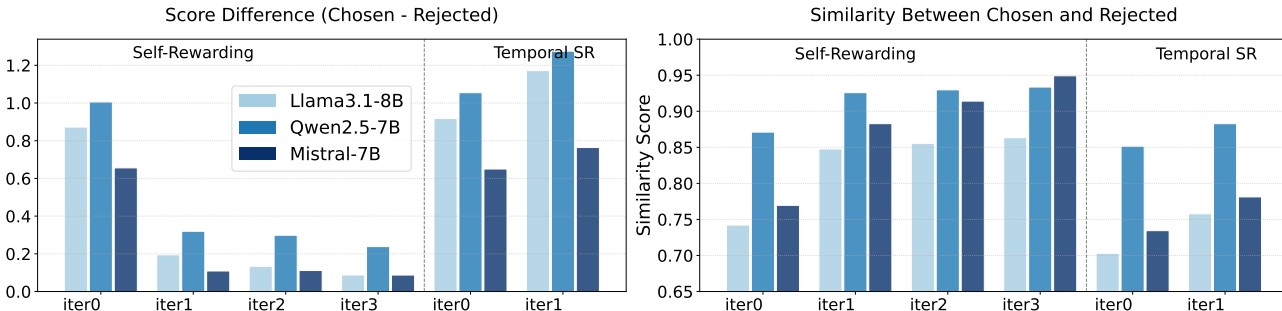

*Figure 1.* Comparison of response preference dynamics between Self-Rewarding and Temporal Self-Rewarding (Temporal SR) frameworks across iterations. We track: (1) the score difference (chosen - rejected) evaluated by GPT-4o-mini (the scoring prompt is the same as that used in Temporal Self-Rewarding, detailed in Appendix A.1) and (2) similarity between chosen and rejected responses(cosin similarity calculated through the last layer's features). With similar computational budgets (4 vs. 2 iterations), Self-Rewarding shows rapid degradation with score gap shrinking 9 times and rapid similarity improvement between chosen and rejected responses of Llama3.1-8B, indicating a progressive narrowing of the quality gap between chosen and rejected samples. Our Temporal approach effectively mitigates this quality convergence.

policy model $\pi_\theta^y$ using preference data. The key components are the implicit reward:

$$
\hat{r} = \beta \Bigg( \big( \log \pi_\theta^y(\mathbf{y}_w|x) - \log \pi_\theta^y(\mathbf{y}_l|x) \big)
$$
$$
- \big( \log \pi_{\text{ref}}^y(\mathbf{y}_w|x) - \log \pi_{\text{ref}}^y(\mathbf{y}_l|x) \big) \Bigg), \quad (1)
$$

and the gradient of the DPO loss $\mathcal{L}_{\text{DPO}}$, which takes the explicit form:

$$
\nabla_\theta \mathcal{L}_{\text{DPO}} = -\beta \underbrace{(1 - \sigma(\hat{r}))}_{\text{Adaptive Weighting}}
$$
$$
\times \underbrace{\big( \nabla_\theta \log \pi_\theta^y(\mathbf{y}_w|x) - \nabla_\theta \log \pi_\theta^y(\mathbf{y}_l|x) \big)}_{\text{Directional Guidance}} . \quad (2)
$$

This gradient reveals two crucial mechanisms: *Adaptive Weighting*, which scales updates based on confidence, and *Directional Guidance*, which pushes the policy toward preferred responses.

**Theorem 2.1** (Bound on Directional Guidance). *Let $\pi_\theta$ be a model that generates a response $\mathbf{y}$ via a latent representation $\mathbf{h}$. Assume the gradient of the log-likelihood, $\nabla_\theta \log \pi_\theta^h(\mathbf{h}|x)$, is $K$-Lipschitz continuous with respect to the latent representation $\mathbf{h}$. Then, for any chosen-rejected pair $(\mathbf{y}_w, \mathbf{y}_l)$ generated from $(\mathbf{h}_w, \mathbf{h}_l)$, the norm of the DPO directional guidance term is bounded as follows:*

$$
\|\nabla_\theta \log \pi_\theta^y(\mathbf{y}_w|x) - \nabla_\theta \log \pi_\theta^y(\mathbf{y}_l|x)\|
$$
$$
= \|\nabla_\theta \log \pi_\theta^h(\mathbf{h}_w|x) - \nabla_\theta \log \pi_\theta^h(\mathbf{h}_l|x)\| \quad (3)
$$
$$
\leq K \cdot \|\mathbf{h}_w - \mathbf{h}_l\|.
$$

We provide the formal proof in Appendix A.2 and offer the following interpretations and insights:

(i) As quantified in Figure 1, representations of chosen $\mathbf{h}_w$ and rejected $\mathbf{h}_l$ responses progressively converge in the self-rewarding paradigm ($\|\mathbf{h}_w - \mathbf{h}_l\| \to 0$). Our theory proves this representational collapse directly causes the gradient difference between the pair to vanish: $\|\nabla_\theta \log \pi_\theta^y(\mathbf{y}_w|x) - \nabla_\theta \log \pi_\theta^y(\mathbf{y}_l|x)\| \to 0$.

(ii) Consequently, in the DPO gradient formulation (Eq. 2), the *Directional Guidance* term approaches zero. Since the *Adaptive Weighting* term is bounded in $(0, 1)$, the overall DPO gradient vanishes ($\nabla_\theta \mathcal{L}_{\text{DPO}} \to 0$), leading to the collapse of the self-rewarding training process.

(iii) By contrast, our framework preserves clear representational difference via two sequential phases—Anchored Rejection and Future-Guided Chosen. This prevents the DPO gradient from vanishing and ensures stable iterative optimization. The details of our proof is in Appendix A.2. We now present the technical details of our approach and the pseudo-code is provided in Algorithm 1.

**SFT Model Initialization**  The initialization process begins with a pretrained foundation model $\mathcal{M}_b$, which we enhance through supervised fine-tuning to develop dual capabilities in response generation and quality assessment. Following Self-rewarding (Yuan et al., 2024), we fine-tune $\mathcal{M}_b$ on two complementary datasets: (1) instruction following fine-tuning (IFT) to improve response generation, and (2) evaluation fine-tuning (EFT) to develop quality assessment capabilities. The resulting model $\mathcal{M}_0$ serves as the foundation for subsequent optimization rounds and provides the anchored rejection responses required by our framework. This initialization process is formally defined as:

$$
\mathcal{M}_0 = \text{SFT}(\mathcal{M}_b, \text{IFT} \cup \text{EFT}). \quad (4)
$$

**Iterative Optimization Process** For each iteration $i$ from 0 to $N$, our framework executes two key phases using the optimization dataset $\mathcal{Q} = \{p_0, \ldots, p_N\}$ containing $Q$ ($Q = $ 5k) queries per prompt set $p_i$.

**Phase 1: Anchored Rejection** For each prompt $p \in p_i$, we generate $K$ responses ($K = 7$) from both the current model $\mathcal{M}_i$ and the initial model $\mathcal{M}_0$, denoted as $r_i = \{r_i^1, \ldots, r_i^K\}$ and $r_0 = \{r_0^1, \ldots, r_0^K\}$ respectively. The current model $\mathcal{M}_i$ scores all responses, producing $s_i = \{s_i^1, \ldots, s_i^K\}$ for its own generations and $s_0 = \{s_0^1, \ldots, s_0^K\}$ for $\mathcal{M}_0$'s outputs.

The chosen response is selected as $r_i^{\arg\max s_i}$ (highest-scoring from $\mathcal{M}_i$), while the rejected response is determined by comparing minima: $r_0^{\arg\min s_0}$ if $\min(s_0) < \min(s_i)$, otherwise $r_i^{\arg\min s_i}$. Valid preference pairs $(chosen, rejected)$ where the chosen score exceeds the rejected score are added to dataset $\mathcal{D}_1$.

We then train a temporary future model $\mathcal{M}_f$ using these anchored rejection pairs:

$$\mathcal{M}_f = \text{DPO}(\mathcal{M}_i, \mathcal{D}_1). \tag{5}$$

**Phase 2: Future-Guided Chosen** For each prompt $p \in p_i$, we generate $K$ responses $r_f = \{r_f^1, \ldots, r_f^K\}$ using $\mathcal{M}_f$ and score them with $\mathcal{M}_i$ to obtain $s_f = \{s_f^1, \ldots, s_f^K\}$. The chosen response is upgraded to $r_f^{\arg\max s_f}$ if $\max(s_f) > \max(s_i)$, otherwise retaining $r_i^{\arg\max s_i}$. These are paired with the same rejected responses from Phase 1 of prompt $p$, with valid pairs added to $\mathcal{D}_2$.

The final model for the next iteration is obtained by:

$$\mathcal{M}_{i+1} = \text{DPO}(\mathcal{M}_i, \mathcal{D}_2). \tag{6}$$

This two-phase process maintains a clear quality gap between chosen and rejected responses by anchoring negatives to past model capabilities while proactively incorporating superior generations from future model predictions. To ensure computationally fair comparisons with self-rewarding approaches, we limit our optimization to two iterations - each requiring training of an additional future model $\mathcal{M}_f$. Despite running for fewer iterations (2 vs. 4 in self-rewarding), our temporal approach achieves superior performance improvements through more effective preference learning.

## 3. Experiments

We conduct extensive experiments using multiple models from the LLaMA (Touvron et al., 2023), Qwen (Yang et al., 2024), and Mistral (Jiang et al., 2023) families as our base models. vLLM (Kwon et al., 2023) is used for inference and deepspeed (Aminabadi et al., 2022) for SFT and DPO.

---

**Algorithm 1** Temporal Self-Rewarding Language Models: Decoupling Chosen-Rejected via Past-Future.

---

1: **Input:** Instruction Fine-Tuning data (IFT), Evaluation Fine-Tuning data (EFT), base model $\mathcal{M}_b$, Iteration Data $\mathcal{Q} = \{p_0, \ldots, p_N\}$ (each $p_i$ contains $Q$ queries).
2: **Output:** Aligned Model $\mathcal{M}_i$ after each iteration $i$ ($0 <= i <= N$).
3: $\mathcal{M}_0 \leftarrow \text{SFT}(\mathcal{M}_b, \text{IFT} + \text{EFT})$ {Supervised Tuning}
4: **for** $i = 0$ **to** $N$ **do**
5:     $\mathcal{D}_1, \mathcal{D}_2 \leftarrow \varnothing$. {Preference datasets for DPO}
6:     **Phase 1:** Decoupling Rejected Responses via $\mathcal{M}_0$
7:     **for** $p \in p_i$ **do**
8:         Generate $K$ responses each from $\mathcal{M}_i$ ($r_i = r_i^1, \ldots, r_i^K$) and $\mathcal{M}_0$ ($r_0 = r_0^1, \ldots, r_0^K$). Then, score all responses using $\mathcal{M}_i$, yielding $s_i = s_i^1, \ldots, s_i^K$ (for $r_i$) and $s_0 = s_0^1, \ldots, s_0^K$ (for $r_0$)
9:         $chosen \leftarrow r_i^{\arg\max s_i}$ {Highest from $\mathcal{M}_i$}
10:         $rejected \leftarrow r_0^{\arg\min s_0}$ if $\min(s_0) < \min(s_i)$ else $r_i^{\arg\min s_i}$ {Lowest from $\mathcal{M}_0$ and $\mathcal{M}_i$}
11:         If $s_{\text{chosen}} > s_{\text{rejected}}$, add $(chosen, rejected)$ to $\mathcal{D}_1$
12:     **end for**
13:     $\mathcal{M}_f \leftarrow \text{DPO}(\mathcal{M}_i, \mathcal{D}_1)$ {Train future model}
14:     **Phase 2:** Decoupling Chosen Responses via $\mathcal{M}_f$
15:     **for** $p \in p_i$ **do**
16:         Generate $K$ responses $r_f = \{r_f^1, \ldots, r_f^K\}$ using $\mathcal{M}_f$ and Score responses: $s_f = \{s_f^1, \ldots, s_f^K\}$ (judged by $\mathcal{M}_i$)
17:         $chosen \leftarrow r_f^{\arg\max s_f}$ if $\max(s_f) > \max(s_i)$ else $r_i^{\arg\max s_i}$ {Highest from $\mathcal{M}_i$ and $\mathcal{M}_f$}
18:         Use same $rejected$ from Phase 1 for this $p$
19:         If $s_{\text{chosen}} > s_{\text{rejected}}$, add $(chosen, rejected)$ to $\mathcal{D}_2$
20:     **end for**
21:     $\mathcal{M}_{i+1} \leftarrow \text{DPO}(\mathcal{M}_i, \mathcal{D}_2)$
22: **end for**

---

The inference and training details are shown in Appendix A.3 and Appendix A.4.

### 3.1. Data Preparation

Following Self-Rewarding, our study processes two primary datasets for model development: the Open Assistant dataset (Köpf et al., 2023) containing question-answer pairs with human judgments (rank 0-4), and the UltraFeedback dataset (Cui et al., 2024) with scored responses. Both datasets provide questions, answers, and rankings but lack scoring explanations. We create three specialized subsets through the following pipeline.

**Instruction Fine-Tuning (IFT) Seed Data** Following (Li et al., 2023b), we construct the IFT seed dataset by sampling high-quality initial conversational turns in English. The preparation process consists of three steps:

First, we extract all English samples from the Open Assistant dataset (oasst1 and oasst2), resulting in 5,000 samples after removing null-score entries. Next, we apply a two-stage selection: (1) identifying highest-ranked responses in initial conversation rounds, and (2) combining these with the top-scoring 25,000 samples from UltraFeedback (excluding the 2,000 most variable entries). The final IFT dataset comprises 5,000 randomly selected question-answer pairs from this collection after thorough shuffling. Each IFT sample contains both the question and its corresponding answer.

**Evaluation Fine-Tuning (EFT) Seed Data** For EFT data preparation, we begin with the 2,000 most variable examples from the UltraFeedback dataset. Each sample's four responses are evaluated by GPT-4o, retaining only those where the model's scoring order matches the original human ratings. This quality control process yields 1,871 validated samples. Importantly, the EFT dataset not only contains questions and answers, but also includes our carefully constructed judge explanations that justify the rankings.

**Iteration Optimization Data** After excluding the 5,000 IFT samples from the mixed dataset, we divide the remaining 20,000 items into five equal parts for iterative optimization. All baseline methods use only questions, except for SPIN, which additionally requires answers as chosen. **Following prior self-rewarding works (typically 2-3 iterations), our Temporal Self-Rewarding conducts 2 iterations, while standard SR is extended to 4 iterations to ensure fair computational comparison.**

### 3.2. Baseline Methods

We conduct comprehensive comparisons of all the following baseline approaches, all of which follow an iterative optimization paradigm. Specifically, Rejection-Sampling SFT performs supervised fine-tuning (SFT) in each round, whereas the other baselines apply direct preference optimization (DPO) iteratively. Their primary differences lie in the strategies for constructing training data.

- **Self-Rewarding**: Using both chosen/rejected samples from current model.

- **Temporal Self-Rewarding (Ours)**: Decoupling Chosen-Rejected via past-future models.

- **Rejection-Sampling SFT** (Liu et al., 2023): Instead of DPO, Fine-tuning with the highest self-rated responses.

- **SPIN** (Chen et al., 2024): Using labels as chosen, current model's responses as rejected.

- **SPIN-Fair**: Serving as a variant of SPIN to ensure fair comparison with other baselines which retains the la-

bel answers as chosen, but selects the lowest-scored responses from model-generated candidates as rejected.

### 3.3. Evaluation Metrics

We evaluate all methods using three widely adopted benchmarks: **AlpacaEval 2.0**, **Arena-Hard-v0.1**, and **MT-Bench**. We adopt **GPT-4o** as the judge model across all benchmarks for its faster inference and lower cost(OpenAI, 2023).

MT-Bench evaluates models' multi-turn dialogue abilities through direct scoring, where the judge assigns numerical ratings to responses for each turn. In contrast, both AlpacaEval 2.0 and Arena-Hard-v0.1 use pairwise comparison to evaluate models. GPT-4o acts as a judge, comparing each model's responses against the baseline. The baselines differ between benchmarks: AlpacaEval 2.0 uses responses from GPT-4 Preview, while Arena-Hard-v0.1 uses GPT-4-0314. AlpacaEval 2.0 employs two primary metrics: win rate and length-controlled win rate. Arena-Hard-v0.1 utilizes score as its evaluation metric to measure win rate. MT-Bench provides first-turn, second-turn and average score.

### 3.4. Main Results

Our experimental evaluation demonstrates significant improvements achieved by Temporal Self-Rewarding across three major benchmarks (AlpacaEval 2.0, Arena-Hard-v0.1, and MT-Bench) compared to existing approaches. Table 1 presents comprehensive results on Llama3.1-8B, comparing our method against four baselines: Self-Rewarding (SR), Rejection-Sampling SFT, SPIN, and SPIN-Fair.

- **Superiority of Self-Rewarding Paradigm:** The self-rewarding approach outperforms traditional methods, with the best iteration achieving 19.69% win rate on AlpacaEval 2.0 compared to 7.20% for SPIN-Fair and 7.33% for Rejection-Sampling. This advantage stems from two key factors: (1) Unlike SPIN's fixed chosen responses (from human data), self-rewarding adopts both chosen and rejected samples to improve, enabling higher performance ceilings. (2) Compared to Rejection-Sampling which only uses positive examples, self-rewarding's preference learning provides more effective optimization signals.

- **Temporal SR Outperforms Self-Rewarding:** Our method achieves consistent improvements over standard self-rewarding across all benchmarks, with particularly notable gains on AlpacaEval 2.0 (29.44% vs 19.69% win rate) and Arena-Hard-v0.1 (14.6% vs 9.4% score). This enhancement results from our temporal coordination strategy which maintains clear quality gaps between chosen and rejected responses - preserving effective learning signals that would otherwise diminish in standard self-rewarding.

*Table 1.* Main results of all baselines of Llama3.1-8B on AlpacaEval 2.0, Arena-Hard-v0.1 and MT-Bench. The best results of each baseline are in bold. The marker $^\dagger$ represents the best results of all baselines.

| Method | Iter | AlpacaEval 2.0 | | | Arena-Hard-v0.1 | | | MT-Bench | | |
|---|---|---|---|---|---|---|---|---|---|---|
| | | LC Win(%) | Win(%) | Length | Score(%) | 95% CI | Length | 1st | 2nd | Avg |
| SFT Model | - | 8.73 | 5.96 | 1324 | 6.3 | (-1.0, 1.0) | 652 | 5.84 | 3.79 | 4.81 |
| Rejection Sampling | 0 | 9.04 | 6.71 | 1385 | 5.4 | (-0.9, 0.9) | 656 | 6.11 | 4.04 | **5.08** |
| | 1 | **9.58** | **7.33** | 1406 | **6.6** | (-1.0, 0.9) | 638 | 6.00 | 4.04 | 5.01 |
| | 2 | 8.82 | 6.96 | 1451 | 4.6 | (-0.8, 1.1) | 606 | 5.99 | 4.14 | 5.06 |
| | 3 | 6.63 | 5.59 | 1487 | 4.5 | (-1.0, 0.8) | 610 | 5.68 | 3.29 | 4.48 |
| SPIN | 0 | 5.63 | 5.28 | 1712 | 3.7 | (-0.8, 0.7) | 823 | 5.61 | 3.75 | 4.68 |
| | 1 | 7.09 | 4.72 | 1150 | 3.8 | (-0.8, 1.0) | 573 | 5.78 | 4.24 | 5.01 |
| | 2 | 4.97 | 4.72 | 1705 | 3.5 | (-0.8, 0.7) | 962 | 5.64 | 3.89 | 4.76 |
| | 3 | **8.93** | **6.83** | 1404 | **4.7** | (-0.7, 0.8) | 666 | 5.96 | 4.26 | **5.11** |
| SPIN-Fair | 0 | 7.37 | 6.40 | 1555 | 4.3 | (-0.7, 0.8) | 779 | 5.60 | 4.08 | 4.84 |
| | 1 | 7.82 | 5.59 | 1239 | 3.4 | (-0.7, 0.7) | 601 | 6.02 | 4.22 | **5.09** |
| | 2 | 5.83 | 5.47 | 1736 | 2.9 | (-0.7, 0.8) | 1000 | 5.65 | 3.85 | 4.75 |
| | 3 | **9.82** | **7.20** | 1398 | **4.7** | (-0.9, 1.0) | 622 | 5.89 | 4.29 | **5.09** |
| Self-Rewarding (SR) | 0 | 13.29 | 10.99 | 1567 | 6.7 | (-1.2, 1.0) | 578 | 6.19 | 4.46 | 5.33 |
| | 1 | 17.00 | 15.71 | 1789 | 7.7 | (-1.4, 1.7) | 592 | 6.50 | 4.70 | 5.60 |
| | 2 | 17.54 | 17.08 | 1865 | **9.4** | (-1.3, 1.4) | 592 | 6.60 | 4.89 | **5.74** |
| | 3 | **19.92** | **19.69** | 1882 | 8.8 | (-1.3, 1.3) | 613 | 6.66 | 4.66 | 5.66 |
| Temporal SR (Ours) | 0 | 20.48 | 19.07 | 1820 | 11.3 | (-1.8, 1.3) | 605 | 6.61 | 4.98 | 5.79 |
| | 1 | **27.94**$^\dagger$ | **29.44**$^\dagger$ | 2063 | **14.6**$^\dagger$ | (-1.7, 1.7) | 698 | 6.84 | 4.94 | **5.89**$^\dagger$ |

- **Iteration Dynamics and Practical Implications:** We observe varying optimal iteration points across methods and benchmarks. While self-rewarding requires 4 iterations to peak, our Temporal SR achieves best performance at iteration 1 on Arena-Hard-v0.1 and iteration 0 on MT-Bench. This variation suggests practitioners should carefully monitor performance across iterations to avoid overfitting (as seen in Rejection-Sampling's performance decline after iteration 1).

The results validate our key insight: decoupling chosen and rejected responses through temporal coordination (past anchoring and future guidance) sustains effective preference learning signals. Additional ablation studies in Section 3.5.1 further analyze the contributions of each component.

### 3.5. Ablation Studies

We conduct comprehensive ablation studies to examine the effectiveness of our Temporal Self-Rewarding mechanism and its key components.

#### 3.5.1. Past-Future Model Ablation

To investigate the impact of the Past and Future models on our method, we conduct an ablation study by selectively removing each component to optimize the model with only one side. This setup allows us to directly compare the optimization effects of the Past component on rejected examples and the Future model on chosen examples. As shown in Table 2, the Past model achieves significant improvements over

the Self-Rewarding baseline, with substantial gains across all metrics. Conversely, when retaining only the Future model (Temporal SR w/o Past), we also observe substantial gains over the Self-Rewarding baseline, where ArenaHard jumps from 6.7 to 14.0 and LC Win rises from 13.29 to 23.31. This indicates that refining chosen examples via the Future model provides a strong learning signal in the early stage of optimization. However, its improvements saturate quickly across iterations (e.g., Win Rate slightly drops from 22.05 to 21.49 at iter 1), suggesting that the headroom for sharpening chosen responses is inherently limited once they already receive high reward scores. Taken together, it is not surprising that the Past model plays a more sustained role than the Future model: as the policy improves through iterative optimization, generated responses tend to receive consistently high scores, leaving little room for the Future model to further sharpen chosen examples. In contrast, refining rejected examples via the Past model continuously accentuates the contrast between chosen and rejected samples, yielding compounding gains across iterations.

#### 3.5.2. Judge Model Ablation

Self-Rewarding paradigm involves the model generating and scoring its own outputs. Considering that the data used in the DPO process primarily aim to enhance the model's generation capabilities, we conduct an ablation study on the judge model component. Specifically, we employ an off-the-shelf external model AutoJ (in 6B and 13B variants), to score the responses throughout the entire Self-Rewarding and Temporal Self-Rewarding workflows, enabling a comparison of the

*Table 2.* Ablation study of Temporal Self-Rewarding (Temporal SR) components. Metrics include Length Control Win Rate (LC Win), Win Rate (Win), ArenaHard Score, and MT-Bench Average.

| Method | Iter | AlpacaEval 2.0 | | ArenaHard | MT-Bench |
|---|---|---|---|---|---|
| | | LC Win | Win | Score | Average |
| SFT | - | 8.73 | 5.96 | 6.3 | 4.81 |
| Temporal SR w/o Future&Past (Self-rewarding) | 0 | 13.29 | 10.99 | 6.7 | 5.33 |
| | 1 | 17.00 | 15.71 | 7.7 | 5.60 |
| | 2 | 17.54 | 17.08 | **9.4** | **5.74** |
| | 3 | **19.92** | **19.69** | 8.8 | 5.66 |
| Temporal SR w/o Future | 0 | 14.35 | 11.61 | 8.1 | 5.39 |
| | 1 | 20.96 | 19.69 | 10.2 | 5.76 |
| | 2 | 24.75 | 27.20 | 11.4 | 5.86 |
| | 3 | **25.73** | **29.06** | **13.4** | **5.88** |
| Temporal SR w/o Past | 0 | 23.31 | **22.05** | **14.0** | **5.82** |
| | 1 | **24.67** | 21.49 | 13.9 | 5.81 |
| Temporal SR | 0 | 24.08 | 19.07 | 11.3 | 5.79 |
| | 1 | **27.94** | **29.44** | **14.6** | **5.89** |

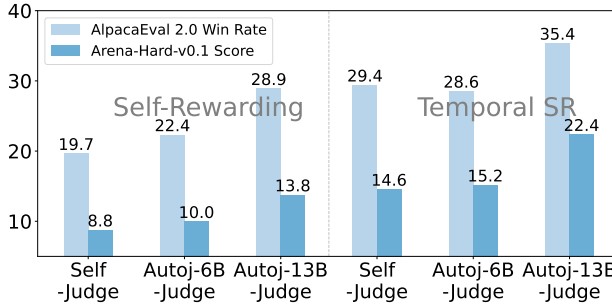

*Figure 2.* The performance of Self-Rewarding and Temporal Self-Rewarding(Temporal SR) using different Judge, evaluated by AlpacaEval 2.0 Win Rate and Arena-Hard-v0.1 Score. This figure illustrate the best model of all iterations in each baseline, detailed results of all iterations can be seen in Appendix A.5.

model improvement achieved by each method. Additionally, we assess whether our method could still outperform Self-Rewarding under different Judge model variants.

As shown in Figure 2, when using AutoJ-6B as the judge, the model's performance under both Self-Rewarding and Temporal Self-Rewarding baselines is overall comparable to that achieved with the Self-Judge approach. However, when AutoJ-13B is used as the judge, both baselines show significant improvements over the Self-Judge method. This suggests that the self-judgment mechanism in Self-Rewarding may lack advantages when faced with a stronger Reward Model. Additionally, it is clear that regardless of whether Self-Judge or AutoJ-Judge is used, Temporal Self-Rewarding consistently outperforms Self-Rewarding, emphasizing the effectiveness of our method regardless of the choice of judge model.

**Generalization Experiment on Model Families** To evaluate the generalization capability of our approach across different model architectures, we test it on Qwen2.5-7B, Llama3.1-8B and Mistral-7B, which differ significantly in design and training methodologies. As shown in Table 3,

*Table 3.* Comparison of different models and their variants using AlpacaEval 2.0 Length Control Win Rate (LC Win), Win Rate (Win), ArenaHard Score, and MT-Bench Average metrics. SR stands for Self-Rewarding and TSR stands for Temporal Self-Rewarding. This table illustrate the best model of all iterations in each baseline, detailed results of all iterations can be seen in Appendix A.6.

| Model | Method | AlpacaEval 2.0 | | ArenaHard | MT-Bench |
|---|---|---|---|---|---|
| | | LC Win | Win | Score | Average |
| Llama3B | SFT | 2.99 | 2.86 | 1.2 | 4.06 |
| | SR | 3.37 | 3.42 | 2.3 | 4.03 |
| | TSR | **3.41** | **8.20** | **2.9** | **4.32** |
| Llama8B | SFT | 8.73 | 5.96 | 6.3 | 4.81 |
| | SR | 19.92 | 19.69 | 8.8 | 5.66 |
| | TSR | **27.94** | **29.44** | **14.6** | **5.89** |
| Llama70B | SFT | 19.96 | 12.80 | 13.0 | 6.06 |
| | SR | 35.57 | 32.91 | 38.9 | 6.93 |
| | TSR | **38.70** | **33.66** | **40.1** | **6.98** |
| Qwen7B | SFT | 11.45 | 7.70 | 12.7 | 5.51 |
| | SR | 21.53 | 18.14 | 21.5 | 6.09 |
| | TSR | **34.01** | **35.90** | **34.4** | **6.29** |
| Mistral7B | SFT | 12.72 | 8.45 | 6.3 | 5.28 |
| | SR | 25.48 | 27.58 | 12.8 | **5.68** |
| | TSR | **30.58** | **35.16** | **15.7** | 5.49 |

Temporal Self-Rewarding consistently outperforms Self-Rewarding and the fine-tuning baseline (SFT) across all tested models. To demonstrate the models' performance across different task categories, we also evaluated Qwen and Mistral on specific scores for each category in the AlpacaEval 2.0 benchmark. Detailed results can be found in Appendix A.7.

**Generalization Experiment on Model Sizes** To further explore the scalability of our approach across different model sizes, we compare Temporal Self-Rewarding and Self-Rewarding on models ranging from small-scale architectures like Llama3.2-3B to mid-scale ones like Llama3.1-8B, and finally to large-scale models such as Llama3.1-70B. As shown in Table 3, Temporal Self-Rewarding consistently delivers superior performance across all model sizes. These results highlight the robustness of Temporal Self-Rewarding not only across diverse model structures but also across different model size.

### 3.6. Extension to Online Reinforcement Learning.

To further validate the generality of our temporal coordination strategy, we draw inspiration from LUFFY (Yan et al., 2025) and extend our offline temporal decoupling framework to the online reinforcement learning setting. Specifically, we augment response groups with offline references generated by both past and future model versions within the Group Relative Policy Optimization (GRPO) framework. For this experiment, we adopt the configuration established by LUFFY, utilizing Qwen-2.5-7B-Math as the base model and `OpenR1-Math-220k` as the training dataset. The validation pool is constructed from multiple mathemati-

cal reasoning benchmarks, including `Olympiad_Bench`, `Minerva`, `Math-500`, `AMC`, `AIME25`, and `AIME24`. As shown in Table 4, our GRPO-Temporal model consistently outperforms the baseline GRPO model across nearly all validation benchmarks, demonstrating that temporal decoupling also enhances online reinforcement learning by stabilizing training dynamics and preserving learning signals throughout iterative updates. Further training details and analyses are provided in Appendix A.8.

*Table 4.* Evaluation results on mathematical reasoning benchmarks, comparing GRPO with GRPO-Temporal (ours). The temporal strategy enhances performance across most tasks by incorporating offline references from past and future models.

| Method | Minerva | Math-500 | AMC | AIME25 | AIME24 | Olympiad_Bench |
|---|---|---|---|---|---|---|
| GRPO | 31.3 | 65.6 | 52.9 | 8.5 | 12.8 | 40.7 |
| GRPO-Temporal | 34.9 | 66.2 | 54.8 | 10.8 | 13.8 | 40.7 |

### 3.7. Out-of-distribution Analysis

While Self-Rewarding and Temporal Self-Rewarding primarily focus on improving instruct-following performance, we also evaluat them on common NLP tasks such as scientific reasoning task(ARC-Challenge), mathematical reasoning problems(GSM8K), factual question answering benchmark(TruthfulQA), and code generation task(HumanEval). Surprisingly, our method achieved impressive results even on these out-of-distribution tasks as illustrated in Table 5. For example, on the reasoning-heavy GSM8K dataset, Temporal Self-Rewarding (iter1) significantly improved accuracy from 0.5299 under SFT to 0.5625, outperforming Self-Rewarding by a substantial margin. The same improvement can also been vitnessed on HumanEval.

*Table 5.* Evaluation results on some NLP Benckmarks. The base model of all baselines is Llama3.1-8B. SR stands for Self-Rewarding.

| Method | ARC | GSM8K | TruthfulQA | HumanEval |
|---|---|---|---|---|
| SFT | 0.5307 | 0.5299 | 0.5049 | 0.2195 |
| SR Iter0 | 0.5384 | 0.5322 | 0.5164 | 0.2317 |
| SR Iter1 | 0.5410 | 0.5459 | 0.5181 | 0.2378 |
| SR Iter2 | 0.5392 | 0.5489 | 0.5187 | 0.2378 |
| SR Iter3 | 0.5375 | 0.5497 | 0.5177 | 0.2378 |
| Temporal SR Iter0 | 0.5452 | 0.5588 | 0.5374 | 0.2439 |
| Temporal SR Iter1 | **0.5495** | **0.5625** | **0.5443** | **0.2622** |

## 4. Related Work

**Self-Rewarding Language Models** Recent advances in self-rewarding language models have demonstrated practically feasible promising alternatives to traditional human-supervised approaches. The foundational work by Yuan et al. (Yuan et al., 2024) first proposed the concept of models serving dual roles as both generators and evaluators, establishing an iterative optimization cycle that combines generation and self-assessment. Subsequent research has focused on enhancing the judging capabilities within this paradigm, with Meta-Rewarding (Wu et al., 2024) introducing self-evaluation mechanisms to refine judgment skills. The field has also seen innovations in consistency regularization for reward models (Wang et al., 2024b) and self-consistency mechanisms for internal rewards (Zhou et al., 2025), particularly in specialized domains like mathematical reasoning (Zhang et al., 2025). These developments collectively represent a shift from static reward models to dynamic, co-evolving generation and evaluation frameworks, though they share the common limitation of inherently synchronized quality improvement between chosen and rejected samples that our work addresses.

**Preference Learning and Response Diversity** The relationship between reinforcement learning and response diversity has been extensively studied in language model alignment. Zhang et al. (Zhang et al., 2024) and Kirk et al. (Kirk et al., 2023) documented the phenomenon of reduced diversity post-reinforcement learning, while Razin et al. (Razin et al., 2025) analyzed the theoretical foundations of effective preference learning. Direct Preference Optimization (DPO) (Rafailov et al., 2023) emerged as a significant advancement over traditional reinforcement learning from human feedback (RLHF), with subsequent variants like DVPO (Lanchantin et al., 2025) exploring mechanisms to maintain meaningful quality gaps between samples. Our work builds upon these insights by introducing temporal decoupling of sample quality levels, addressing diminishing contrast between chosen and rejected responses that has been observed in iterative optimization processes.

## 5. Limitations

While our Temporal Self-Rewarding approach demonstrates significant improvements over conventional self-rewarding methods, several limitations warrant discussion. First, the computational overhead of our method is approximately twice that of standard self-rewarding due to the additional future model training in each iteration. Although we maintain computational fairness caused by model training through reducing the number of iterations (2 of ours vs. 4 in self-rewarding), the per-iteration cost remains higher for multiple generations of past, present and future models.

Second, while our framework is theoretically compatible with judge optimization techniques in self-rewarding paradigm such as meta-rewarding, we are unable to explore this integration due to limitations in time and research resources. We believe this represents a promising direction for future work that could improve the system's performance.

# 6. Conclusion

In this paper, we introduced Temporal Self-Rewarding Language Models, a novel framework that addresses the critical limitation of diminishing preference signals in conventional self-rewarding approaches. Our method strategically coordinates past, present, and future model generations through two key innovations: Anchored Rejection that fixes negative samples using past model outputs, and Future-Guided Chosen that incorporates next-generation model predictions to maintain quality differentiation. Extensive experiments across three model families (Llama, Qwen, Mistral) and multiple benchmarks (AlpacaEval 2.0, Arena-Hard, MT-Bench) demonstrate that our approach significantly outperforms standard self-rewarding methods while requiring fewer iterations. The success of Temporal Self-Rewarding establishes a new paradigm for iterative self-improvement that preserves effective learning signals through temporal decoupling of sample quality levels, offering both theoretical insights and practical advancements in LLM alignment.

# Impact Statement

This paper presents work whose goal is to advance the field of Machine Learning. There are many potential societal consequences of our work, none which we feel must be specifically highlighted here.

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

# A. Appendices

## A.1. Judging prompt

As illustrated in Figure 3, the user prompt is strategically constructed following the self-rewarding paradigm. Additionally, we construct system prompt carefully as Figure 4.

Review the user's question and the corresponding response using the additive 5-point scoring system described below. Points are accumulated based on the satisfaction of each criterion:

- Add 1 point if the response is relevant and provides some information related to the user's inquiry, even if it is incomplete or contains some irrelevant content.
- Add another point if the response addresses a substantial portion of the user's question, but does not completely resolve the query or provide a direct answer.
- Award a third point if the response answers the basic elements of the user's question in a useful way, regardless of whether it seems to have been written by an AI Assistant or if it has elements typically found in blogs or search results.
- Grant a fourth point if the response is clearly written from an AI Assistant's perspective, addressing the user's question directly and comprehensively, and is well-organized and helpful, even if there is slight room for improvement in clarity, conciseness or focus.
- Bestow a fifth point for a response that is impeccably tailored to the user's question by an AI Assistant, without extraneous information, reflecting expert knowledge, and demonstrating a high-quality, engaging, and insightful answer.

User: {question}

<response> {response} </response>

After examining the user's instruction and the response:

- Briefly justify your total score, up to 100 words.
- Conclude with the score using the format: "Score: <total points>"

Remember to assess from the AI Assistant perspective, utilizing web search knowledge as necessary. To evaluate the response in alignment with this additive scoring model, we'll systematically attribute points based on the outlined criteria.

*Figure 3.* User Prompt of Judging in Self-Rewarding paradigm.

## A.2. DPO Loss Function and Gradient Derivation

The foundation of our analysis is the DPO loss function, which aims to directly optimize a policy model on a static dataset of preferences.

**The DPO Objective**   The DPO loss is defined as:

$$L_{\text{DPO}}(\pi_\theta; \pi_{\text{ref}}) = -\mathbb{E}_{(x,y_w,y_l)\sim D}[\log \sigma(\hat{r})],$$

where $\hat{r}$ represents the reward difference between the policy model $\pi_\theta$ and a reference model $\pi_{\text{ref}}$:

$$\hat{r} = \beta\left(\log \frac{\pi_\theta(y_w|x)}{\pi_{\text{ref}}(y_w|x)} - \log \frac{\pi_\theta(y_l|x)}{\pi_{\text{ref}}(y_l|x)}\right). \tag{7}$$

> You are a helpful and precise evaluator. Provide objective, constructive, and detailed feedback. Be concise, clear, and tailored to the context. Your goal is to support refinement and informed decision-making.

*Figure 4.* System Prompt of Judging in Self-Rewarding paradigm.

Here:

- $\pi_\theta$: The current policy model being trained (e.g., $\mathcal{M}_{i+1}$ in our analysis).

- $\pi_{\text{ref}}$: A fixed reference model (e.g., $\mathcal{M}_i$).

- $y_w$: The preferred or "chosen" response.

- $y_l$: The dispreferred or "rejected" response.

- $\beta$: A temperature parameter scaling the reward.

By rearranging the terms of $\hat{r}$ in (7), the optimization goal could be given more clearly:

$$\hat{r} = \beta[\underbrace{(\log \pi_\theta(y_w|x) - \log \pi_\theta(y_l|x))}_{\text{Implicit reward of policy model}}$$
$$- \underbrace{(\log \pi_{\text{ref}}(y_w|x) - \log \pi_{\text{ref}}(y_l|x))}_{\text{Implicit reward of reference model}}] \tag{8}$$

**Inference** The objective of DPO is to maximize the gap between the policy model's and the reference model's implicit rewards for the preference pair $(y_w, y_l)$. In other words, it encourages the policy $\pi_\theta$ to be better at discriminating between chosen and rejected samples than the reference model $\pi_{\text{ref}}$.

**Gradient Derivation** To understand how the model parameters $\theta$ are updated, we derive the gradient of the loss for a single sample, $L = -\log \sigma(\hat{r})$.

Applying the chain rule, and knowing that the derivative of the sigmoid function $\sigma'(z) = \sigma(z)(1 - \sigma(z))$, we get:

$$\nabla_\theta L = -\frac{1}{\sigma(\hat{r})} \cdot \nabla_\theta \sigma(\hat{r})$$
$$= -\frac{1}{\sigma(\hat{r})} \cdot \sigma(\hat{r})(1 - \sigma(\hat{r})) \cdot \nabla_\theta \hat{r}$$
$$= -(1 - \sigma(\hat{r})) \cdot \nabla_\theta \hat{r}. \tag{9}$$

The gradient of $\hat{r}$ with respect to $\theta$ only depends on the terms involving $\pi_\theta$:

$$\nabla_\theta \hat{r} = \beta(\nabla_\theta \log \pi_\theta(y_w|x) - \nabla_\theta \log \pi_\theta(y_l|x)). \tag{10}$$

Combining these, we arrive at the final gradient expression:

$$\nabla_\theta L_{\text{DPO}} \propto -(1 - \sigma(\hat{r})) \cdot \beta\big(\nabla_\theta \log \pi_\theta(y_w|x)$$
$$-\nabla_\theta \log \pi_\theta(y_l|x)\big). \tag{11}$$

**Inference**: The gradient update consists of two key components:

- **Weighting Term** $(1 - \sigma(\hat{r}))$: This term modulates the magnitude of the update. When the model is uncertain or incorrect (i.e., $\hat{r}$ is small or negative), this weight approaches 1, leading to a large update. When the model is confident and correct ($\hat{r}$ is large), the weight approaches 0, reducing the update.

- **Direction Term** $(\nabla_\theta \log \pi_\theta(y_w) - \nabla_\theta \log \pi_\theta(y_l))$: This term dictates the update direction, effectively increasing the log-probability of the chosen response $y_w$ and decreasing the log-probability of the rejected response $y_l$.

**Applying the Gradient Analysis** We now use this gradient derivation to analyze the dynamics of both standard and temporal self-rewarding methods. In this iterative context, we are training model $\mathcal{M}_{i+1}$ with $\mathcal{M}_i$ as the reference.

### A.2.1. SCENARIO A: STANDARD SELF-REWARDING (THE PROBLEM)

- **Setup**:

  - Policy model $\pi_\theta = \mathcal{M}_{i+1}$
  - Reference model $\pi_{\text{ref}} = \mathcal{M}_i$
  - Both $y_w$ and $y_l$ are generated by the current model, $\mathcal{M}_i$.

- **Analysis**: As the iteration $i$ progresses, the capability of $\mathcal{M}_i$ improves. Consequently, the intrinsic quality of both its best outputs ($y_w$) and its worst outputs ($y_l$) rises. The qualitative gap between them narrows.

  This has a direct impact on $\hat{r}$. As $\mathcal{M}_{i+1}$ begins its training (where $\mathcal{M}_{i+1} \approx \mathcal{M}_i$), the two main terms in Equation 8 become nearly equal:

$$(\log \mathcal{M}_{i+1}(y_w) - \log \mathcal{M}_{i+1}(y_l)) \approx$$
$$(\log \mathcal{M}_i(y_w) - \log \mathcal{M}_i(y_l))$$

  This leads to $\hat{r} \to 0$. As $\hat{r}$ approaches zero, the weighting term $(1 - \sigma(\hat{r}))$ approaches a constant 0.5. More critically, the underlying signal—the distinguishability between $y_w$ and $y_l$—is diminished. The optimization landscape flattens, providing a weak and ambiguous signal for the model, thus impeding further learning.

### A.2.2. SCENARIO B: TEMPORAL SELF-REWARDING (THE SOLUTION)

This method decouples the data generation to counteract signal decay.

- **Setup**:

  - Policy model $\pi_\theta = \mathcal{M}_{i+1}$
  - Reference model $\pi_{\text{ref}} = \mathcal{M}_i$
  - **Anchored Rejection**: $y_l$ is sampled from an early, fixed, and weaker model, $\mathcal{M}_0$.
  - **Future-Guided Chosen**: $y_w$ is sampled from a stronger, temporary future model, $\mathcal{M}_f$.

- **Analysis**:

  1. **Effect of Anchored Rejection**: By anchoring $y_l$ to a consistently low-quality source ($\mathcal{M}_0$), the method prevents the quality of negative samples from inflating. Both the reference model $\mathcal{M}_i$ and the policy model $\mathcal{M}_{i+1}$ can easily assign a very low log-probability to this poor sample. This makes the $-\nabla_\theta \log \pi_\theta(y_l)$ part of the gradient's direction term consistently large and provides a clear "avoid this" signal.
  2. **Effect of Future-Guided Chosen**: The chosen sample $y_w$ comes from a model $\mathcal{M}_f$ that is more capable than the current reference model $\mathcal{M}_i$. The reference model $\mathcal{M}_i$ will likely assign a modest log-probability, $\log \mathcal{M}_i(y_w)$, to this advanced sample. However, the policy model $\mathcal{M}_{i+1}$ is explicitly trained to learn to generate such higher-quality outputs. Its objective is to make $\log \mathcal{M}_{i+1}(y_w)$ very high.

  **Overall Impact on the Gradient**: This "past-future" decoupling artificially preserves and widens the quality gap between $y_w$ and $y_l$. It systematically forces a large difference between the policy model's implicit reward and the reference model's implicit reward, ensuring that the reward signal $\hat{r}$ remains large and positive. Consequently, the gradient (Equation 11) remains strong, clear, and stable throughout the training process, effectively resolving the signal decay problem inherent in the standard self-rewarding approach.

### A.3. Inference details

We use batched inference of vllm to accelerate the generation and judging process. The generation parameters are temperature=1.0, top_p=1.0 and max_token=1024.

*Table 6.* Detailed results of all iterations of Self-Rewarding and Temporal Self-Rewarding(SR and TSR) using different models as judge(iterJudge/autoj-6b/autoj-13b). The best results of each baseline are in bold. The marker † represents the best results of all baselines.

| Method | Iter | AlpacaEval 2.0 | | ArenaHard | MT-Bench |
| | | LC Win | Win | Score | Average |
|---|---|---|---|---|---|
| SFT | - | 8.73 | 5.96 | 6.3 | 4.81 |
| SR (iterJudge) | 0 | 13.29 | 10.99 | 6.7 | 5.33 |
| | 1 | 17.00 | 15.71 | 7.7 | 5.60 |
| | 2 | 17.54 | 17.08 | **9.4** | **5.74** |
| | 3 | **19.92** | **19.69** | 8.8 | 5.66 |
| TSR (iterJudge) | 0 | 20.48 | 19.07 | 11.3 | 5.79 |
| | 1 | 27.94† | 29.44† | 14.6† | 5.89† |
| SR (autoj-6b) | 0 | 16.28 | 14.84 | 8.6 | 5.46 |
| | 1 | 18.15 | 19.13 | 8.0 | 5.52 |
| | 2 | 19.64 | 20.50 | **10.2** | **5.54** |
| | 3 | **20.95** | **22.36** | 10.0 | 5.45 |
| TSR (autoj-6b) | 0 | 22.67† | 24.47 | 10.7 | 5.74 |
| | 1 | 22.30 | 28.57† | 15.2† | 5.83† |
| SR (autoj-13b) | 0 | 14.59 | 14.31 | 9.5 | 5.33 |
| | 1 | 19.42 | 22.80 | 9.8 | 5.55 |
| | 2 | **21.43** | 28.07 | 12.0 | **5.62** |
| | 3 | 19.59 | **28.94** | 13.8 | 5.58 |
| TSR (autoj-13b) | 0 | 26.61† | 31.30 | 14.7 | 5.78† |
| | 1 | 19.87 | 35.40† | 22.4† | 5.69 |

## A.4. Training details

We conduct supervised fine-tuning (SFT) and direct preference optimization (DPO) with DeepSpeed ZeRO-3 optimization for memory-efficient training.

The SFT models are trained for 3 epochs with a learning rate of $2.0 * 10^{-6}$ and a global batch size of 32 (4 per device * 8 GPUs), using cosine learning rate scheduling with 10% warmup ratio.

For DPO training, the models are trained for 1 epoch with a learning rate of $5.0 * 10^{-7}$, $\beta = 0.1$ and a global batch size of 32 (4 per device * 8 GPUs). The training use cosine learning rate scheduling with 10% warmup ratio, maintaining the maximum sequence length of 2048 tokens and with a maximum prompt length of 512 tokens.

## A.5. Details of Judge Model Ablation

We employ an off-the-shelf external model Au toJ (in 6B and 13B variants), to score the responses throughout the entire Self-Rewarding and Temporal Self-Rewarding workflows, enabling a comparison of the model improvement achieved by each method. We demonstrate our detailed results of all iterations in Table 6.

## A.6. Details of Generalization Experiment

To evaluate the generalization capability of our approach across different model architectures and scales, we test it on Qwen2.5-7B, Llama3.1-8B, Mistral-7B, Llama3.2-3B and Llama3.1-70B. We demonstrate our detailed results of all iterations in Table 7.

## A.7. AlpacaEval win rate across categories.

To assess model capabilities across diverse task domains, we conduct a comprehensive evaluation of Qwen2.5-7B and Mistral-7B across different categories from the AlpacaEval 2.0 benchmark. Retaining the optimal iteration models from both Self-Rewarding and Temporal Self-Rewarding methodologies, our analysis reveals that Temporal Self-Rewarding achieves consistent performance improvements over both baseline Self-Rewarding and the initial SFT model in virtually all categories.

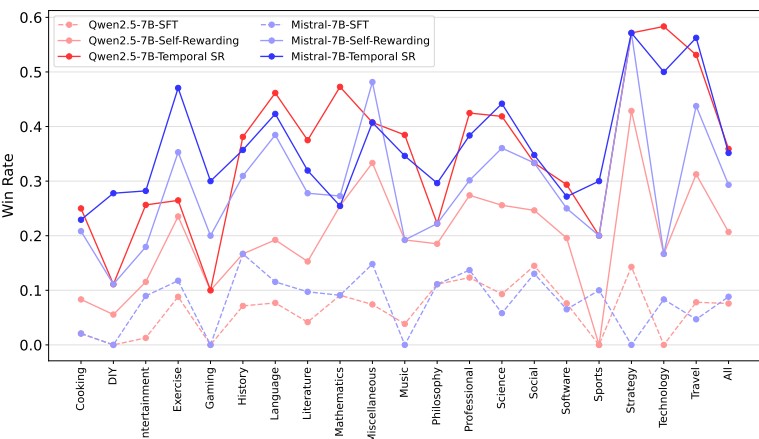

*Figure 5.* AlpacaEval win rate breakdown for instruction categories of Qwen2.5-7B and Mistral-7B. Temporal Self-Rewarding models give gains across nearly all topics than Self-Rewarding and SFT intial.

*Table 7.* Detailed results of all iterations of Self-Rewarding and Temporal Self-Rewarding(SR and TSR) across models of different families and scales. The best results of each baseline are in bold. The marker $^\dagger$ represents the best results of all baselines.

| Model | Method | Iter | AlpacaEval 2.0 LC Win | Win | ArenaHard Score | MT-Bench Average |
|---|---|---|---|---|---|---|
| Llama3B | SFT | - | 2.99 | 2.86 | 1.2 | 4.06 |
| | SR | 0 | 3.93 | 3.85 | 1.8 | 3.91 |
| | | 1 | 2.74 | 2.86 | 1.5 | 3.87 |
| | | 2 | 2.57 | 2.73 | 2.2 | **4.08** |
| | | 3 | **3.37** | **3.42** | **2.3** | 4.03 |
| | TSR | 0 | **4.79**$^\dagger$ | 5.71 | 2.0 | 4.25 |
| | | 1 | 3.41 | **8.20**$^\dagger$ | **2.9**$^\dagger$ | **4.32**$^\dagger$ |
| Llama8B | SFT | - | 8.73 | 5.96 | 6.3 | 4.81 |
| | SR | 0 | 13.29 | 10.99 | 6.7 | 5.33 |
| | | 1 | 17.00 | 15.71 | 7.7 | 5.60 |
| | | 2 | 17.54 | 17.08 | **9.4** | **5.74** |
| | | 3 | **19.92** | **19.69** | 8.8 | 5.66 |
| | TSR | 0 | 20.48 | 19.07 | 11.3 | 5.79 |
| | | 1 | **27.94**$^\dagger$ | **29.44**$^\dagger$ | **14.6**$^\dagger$ | **5.89**$^\dagger$ |
| Llama70B | SFT | - | 19.96 | 12.80 | 13.0 | 6.06 |
| | SR | 0 | 29.42 | 22.92 | 26.2 | 6.66 |
| | | 1 | 33.51 | 28.20 | 29.2 | 6.86 |
| | | 2 | 33.14 | 29.88 | 34.8 | **6.98**$^\dagger$ |
| | | 3 | **35.57** | **32.91** | **38.9** | 6.93 |
| | TSR | 0 | 30.33 | 23.11 | 30.7 | 6.75 |
| | | 1 | **38.70**$^\dagger$ | **33.66**$^\dagger$ | **40.1**$^\dagger$ | **6.98**$^\dagger$ |
| Qwen7B | SFT | - | 11.45 | 7.70 | 12.7 | 5.51 |
| | SR | 0 | 19.82 | 12.92 | 18.4 | 5.93 |
| | | 1 | 21.66 | 15.53 | 19.5 | **6.12** |
| | | 2 | 20.24 | 15.65 | **22.0** | 6.00 |
| | | 3 | **21.53** | **18.14** | 21.5 | 6.09 |
| | TSR | 0 | 27.85 | 24.78 | 27.2 | 6.25 |
| | | 1 | **34.01**$^\dagger$ | **35.90**$^\dagger$ | **34.4**$^\dagger$ | **6.29**$^\dagger$ |
| Mistral7B | SFT | - | 12.72 | 8.45 | 6.3 | 5.28 |
| | SR | 0 | 17.98 | 15.09 | 9.1 | 5.55 |
| | | 1 | 18.97 | 17.70 | 9.8 | 5.35 |
| | | 2 | **26.15** | 24.10 | 9.8 | 5.49 |
| | | 3 | 25.48 | **27.58** | 12.8 | **5.68** |
| | TSR | 0 | **32.11**$^\dagger$ | 32.05 | 14.0 | **5.76**$^\dagger$ |
| | | 1 | 30.58 | **35.16**$^\dagger$ | **15.7**$^\dagger$ | 5.49 |

## A.8. Details of our extended experiment on Online Reinforcement Learning.

Our online reinforcement learning experiment is implemented using the LUFFY framework (Yan et al., 2025), built upon the VERL (Sheng et al., 2024). We perform full fine-tuning of the Qwen-2.5-7B-Math model on 8 GPUs for 100 steps, employing a learning rate of $1 \times 10^{-6}$ and the `OpenR1-Math-220k` (Hugging Face, 2025) dataset.

The construction of offline reference data involves generating solutions from both a past and a future model for each training problem, filtering for correctness, and then randomly selecting one correct solution from one temporal source. This yields a balanced set where half of the references originate from past models and half from future models. In addition to our

*Table 8.* Evaluation results on mathematical reasoning benchmarks, comparing GRPO, GRPO-Past and GRPO-Temporal (ours).

| Method | Minerva | Math-500 | AMC | AIME25 | AIME24 | Olympiad_Bench |
|---|---|---|---|---|---|---|
| GRPO | 31.3 | 65.6 | 52.9 | 8.5 | 12.8 | 40.7 |
| GRPO-Past | 33.5 | 64.6 | 53.4 | 8.9 | 12.8 | 40.3 |
| GRPO-Temporal | 34.9 | 66.2 | 54.8 | 10.8 | 13.8 | 40.7 |

full GRPO-Temporal method, we conduct an ablation study with a variant named **GRPO-Past**, which uses only correct solutions from the past model as offline references. The results for both, presented in Table 8, demonstrate that the balanced, temporally decoupled references in GRPO-Temporal contribute significantly to performance gains over the baseline and the past-only variant.

