# OpenReview forum: "Temporal Self-Rewarding Language Models: Decoupling Chosen-Rejected via Past-Future"
_ICML.cc/2026/Conference — ICML 2026 regular_

### Official Review · Reviewer_xYyV · 2026-03-07

**Soundness:** 3
**Presentation:** 3
**Significance:** 3
**Originality:** 3
**Overall Recommendation:** 4
**Confidence:** 3

**Summary:**

This paper proposes Temporal Self-Rewarding Language Models to address a key limitation of existing self-rewarding approaches: chosen and rejected responses tend to become overly similar over training iterations, which undermines preference learning and risks model collapse. To mitigate this, the method anchors rejection responses to current-generation outputs while pairing them with future-guided chosen responses derived from next-generation model predictions, thereby maintaining a more meaningful and stable preference gap. Experiments across multiple model families, including Llama, Qwen, and Mistral, demonstrate consistent improvements on alignment benchmarks such as AlpacaEval 2.0, Arena-Hard, and MT-Bench. The method is further validated on out-of-distribution tasks and extended to an online reinforcement learning setting.

**Compliance With Llm Reviewing Policy:**

Affirmed.

**Final Justification:**

The paper identifies an important limitation of iterative self-rewarding methods, namely the gradual collapse of preference signals as chosen and rejected responses become increasingly similar, which will undermine the preference training. During rebuttal, the author gives detailed explaination and further experiments on convergence and other method comparison. Hence, I'd like to raise my score to 4.

**Key Questions For Authors:**

1. The experiments only report two iterations for Temporal Self-Rewarding. Do the authors have results for additional iterations (e.g., 3–5)? Self-improving training methods often show different behaviors across iterations, and it would be helpful to understand whether the proposed method eventually converges and at which iteration.

**Limitations:**

yes

**Strengths And Weaknesses:**

**Strengths**
1. The paper identifies an important limitation of iterative self-rewarding methods, namely the gradual collapse of preference signals as chosen and rejected responses become increasingly similar, which will undermine the preference training.
2. The method is evaluated across multiple model families and sizes, suggesting that the approach generalizes reasonably well across architectures.
3. The paper includes comprehensive ablation studies that isolate the effects of the past and future components, which helps clarify the contribution of each module.

**Weaknesses**
1. The proposed method introduces non-trivial computational overhead, as each iteration requires training an auxiliary future model prior to updating the main model.
2. Some concurrent self-improving methods (e.g., DRIFT [1]) achieve competitive results without introducing additional model training within each iteration by only anchoring rejection. In the ablation study (Table 2), most of the performance gains appear to come from the past (anchored rejection) component, while the future-guided chosen mechanism provides only marginal improvements. This raises the question of whether the doubled per-iteration training cost introduced by the temporary future model is truly necessary.
3. Convergence analysis and results beyond two iterations are absent from the paper, which are essential for assessing the long-term stability and reliability of the self-improving method.

Reference:
1. Wang, Yifan, Bolian Li, Junlin Wu, Zhaoxuan Tan, Zheli Liu, Ruqi Zhang, Ananth Grama, and Qingkai Zeng.  DRIFT: Learning from Abundant User Dissatisfaction in Real-World Preference Learning.  *International Conference on Learning Representations (ICLR)*, 2026.

---

> ### Author Rebuttal · Authors · 2026-03-28
>
> We sincerely appreciate your thorough evaluation and respond to your concerns below.
>
> **Reviewer’s Weakness 1**
>
> The proposed method introduces non-trivial computational overhead, as each iteration requires training an auxiliary future model prior to updating the main model.
>
> **Our Response 1**
>
> We acknowledge this valid concern. The introduction of a future model in each iteration indeed adds an additional training step per round, and we have noted this computational overhead in the limitations section of our paper.
>
> To ensure a fair comparison, we designed our experimental setup such that Self-Rewarding runs for **four** iterations while Temporal Self-Rewarding runs for **two** iterations. This configuration ensures that both methods involve a comparable number of model training steps, allowing for a meaningful and equitable comparison. Under this setting, we demonstrate that Temporal Self-Rewarding achieves substantially better performance than the standard Self-Rewarding approach, clearly validating the effectiveness of our method.
>
>
> **Reviewer’s Weakness 2**
>
> Some concurrent self-improving methods (e.g., DRIFT [1]) achieve competitive results without introducing additional model training within each iteration by only anchoring rejection. ... the temporary future model is truly necessary.
>
> **Our Response 2**
>
> We thank the reviewer for bringing DRIFT to our attention. We have carefully read this concurrent work and note that it is indeed a special case of our Temporal Self-Rewarding. **Specifically, it adopts only the past (anchored rejection) component without incorporating the future model. We also observe that DRIFT has cited our Temporal Self-Rewarding work, acknowledging our contribution.**
>
> We believe our architecture is more comprehensive and generalizable than DRIFT, and that the future component also offers significant value—especially in tasks requiring high-quality, definitive answers, such as mathematical reasoning and code generation. In such domains, the future model can explore and generate better chosen responses beyond what the current model alone can find, thereby improving DPO data quality.
>
> To validate this, we conduct additional experiments on GSM8K and HumanEval, as a complement in section 3.7. The results are presented below, where Temporal Self-Rewarding (with past and future components) consistently outperforms DRIFT (which only past component), confirming that the future model provides substantial complementary benefits in such tasks demanding higher-quality answers.
>
> | Method | Iteration | GSM8K | HumanEval |
> |--------|-----------|-------|-----------|
> | SFT | - | 0.5299 | 0.2195 |
> | SR | 0 | 0.5322 | 0.2317 |
> |  | 1 | 0.5459 | 0.2378 |
> |  | 2 | 0.5489 | 0.2378 |
> |  | 3 | 0.5497 | 0.2378 |
> | Temporal SR w/o Future (DRIFT) | 0 | 0.5322 | 0.2378 |
> |  | 1 | 0.5489 | 0.2317 |
> |  | 2 | 0.5489 | 0.2439 |
> |  | 3 | 0.5510 | 0.2473 |
> | Temporal SR | 0 | 0.5588 | 0.2439 |
> |  | 1 | 0.5625 | 0.2622 |
>
>
> **Reviewer’s Weakness 3**
>
> Convergence analysis and results beyond two iterations are absent from the paper, which are essential for assessing the long-term stability and reliability of the self-improving method.
>
> **Our Response 3**
>
> We would first like to clarify that our iteration numbers align with or exceed those in related work (e.g., Self-Rewarding: 2, Meta-Rewarding: 3, SPIN: 4, DRIFT: 2). We ran Self-Rewarding and SPIN for four iterations, and Temporal Self-Rewarding for two, under the same computational budget for fairness.
>
> However, motivated by the reviewer's suggestion, we also conduct additional experiments to analyze convergence behavior. We extend Self-Rewarding for 4 more iterations and Temporal Self-Rewarding for 2 more iteration on the Llama-3.1-8B model. The full results are presented below:
>
>
> | Method | Iter | LC Win | Win | ArenaHard Score | MT-Bench Average |
> |--------|------|--------|-----|-----------------|------------------|
> | Self-Rewarding | 0 | 13.29 | 10.99 | 6.7 | 5.33 |
> |  | 1 | 17.00 | 15.71 | 7.7 | 5.60 |
> |  | 2 | 17.54 | 17.08 | 9.4 | 5.74 |
> |  | 3 | 19.92 | 19.69 | 8.8 | 5.66 |
> |  | 4 | 20.14 | 19.92 | 9.0 | 5.69 |
> |  | 5 | 19.58 | 19.78 | 8.7 | 5.70 |
> |  | 6 | 20.31 | 20.11 | 9.1 | 5.71 |
> |  | 7 | 19.87 | 19.95 | 8.9 | 5.70 |
> | Temporal Self-Rewarding | 0 | 20.48 | 19.07 | 11.3 | 5.79 |
> |  | 1 | 27.94 | 29.44 | 14.6 | 5.89 |
> |  | 2 | 30.03 | 32.21 | 15.9 | 5.94 |
> |  | 3 | 30.45 | 31.88 | 15.7 | 5.95 |
>
> Self-Rewarding begins to plateau after Iter3, with all metrics fluctuating narrowly and yielding no meaningful further gains. Temporal Self-Rewarding, by contrast, improves substantially through Iter2, after which performance stabilizes. Importantly, even at convergence, Temporal Self-Rewarding outperforms Self-Rewarding across all metrics. These results demonstrate that Temporal Self-Rewarding achieves superior performance, further validating its effectiveness as a robust self-improving method.
>
> We hope these explanations address your concerns.

---

> > ### Author Rebuttal · Reviewer_xYyV · 2026-04-03
> >
> > Thanks authors detailed explaination and further experiments, which solved most of my concerns. Hence, I'd like to raise my score

---

> > > ### Author Response · Authors · 2026-04-04
> > >
> > > We sincerely thank the reviewer for the positive reassessment and for raising the score. Your constructive feedback is invaluable in strengthening our work. We will carefully incorporate all suggestions into the final version to further improve the quality of our paper.

---

### Official Review · Reviewer_cm93 · 2026-03-10

**Soundness:** 4
**Presentation:** 4
**Significance:** 3
**Originality:** 3
**Overall Recommendation:** 4
**Confidence:** 3

**Summary:**

The paper finds that as training progresses, the gap between chosen and rejected responses tends to shrink, which weakens the preference learning signal. Therefore, the paper introduces Temporal Self-Rewarding. Rejected responses are anchored to earlier model outputs. Experiments on multiple benchmarks and model families show that Temporal Self-Rewarding improves performance over standard self-rewarding approaches.

**Compliance With Llm Reviewing Policy:**

Affirmed.

**Final Justification:**

Thanks for the authors' response, which fully resolved my concerns.

**Key Questions For Authors:**

See in weaknesses.

**Limitations:**

yes

**Strengths And Weaknesses:**

Strengths:
- Temporal Self-Rewarding addresses a practical issue in self-rewarding pipelines, i.e., the quality gap between chosen and rejected responses gradually shrinks, which weakens the preference learning signal for DPO.
- Temporal Self-Rewarding framework is easy to reuse for a SFT+DPO pipeline.
- The experiments show temporal self-rewarding obtains gains compared to standard self-rewarding and the baselines across diverse benchmarks and model sizes.

Weaknesses:
- The analysis of why temporal decoupling improves preference optimization is missing.
- Training an auxiliary future model is required during each iteration, which increases computational cost and complexity.

---

> ### Author Rebuttal · Authors · 2026-03-28
>
> We sincerely appreciate your thoughtful evaluation and recognizing the strengths of our work. We are grateful for the constructive feedback and have addressed the key concerns below.
>
> **Reviewer’s Weakness 1**
>
> The analysis of why temporal decoupling improves preference optimization is missing.
>
> **Our Response 1**
>
> We thank the reviewer for raising this point. In fact, we have provided a systematic analysis of why temporal decoupling improves preference optimization, which we would like to summarize clearly here:
>
> In Section 2 Methodology (Theorem 2.1) and Appendix A.2, we first present a theoretical proof showing that under the standard Self-Rewarding paradigm, the representational difference between chosen and rejected samples progressively shrinks across iterations. This causes the gradient direction term in the DPO loss to vanish, ultimately leading to the saturation of iterative training gains. This theoretical result mathematically establishes the necessity of decoupling to maintain data difference and keep the effectiveness of DPO.
>
> Then we provide an Empirical Validation. As shown in Figure 1, we provide empirical evidence that directly supports our theory. In the standard self-rewarding method, the score gap between chosen and rejected samples shrinks by 9 times, while their representational similarity increases sharply across iterations. These observations confirm the degradation of the preference learning signal as described in our analysis.
>
> Based on the above understanding, we design the Temporal Self-Rewarding framework to directly address this issue.
>
> Anchored Rejection: We anchor rejected samples to the outputs of the initial (past) model, preventing their quality from "inflating" across iterations.
> Future-Guided Chosen: We leverage predictions from a future model to dynamically select higher-quality chosen samples, ensuring that the quality of chosen responses continues to improve.
> By maintaining the quality gap between chosen and rejected responses in this manner, Temporal Self-Rewarding effectively preserves the preference learning signal throughout iterative training.
>
>
> **Reviewer’s Weakness 2**
>
> Training an auxiliary future model is required during each iteration, which increases computational cost and complexity.
>
> **Our Response 2**
>
> We acknowledge this valid concern. The introduction of a future model in each iteration indeed adds an additional training step per round, and we have noted this computational overhead in the limitations section of our paper.
>
> To ensure a fair comparison, we designed our experimental setup such that Self-Rewarding runs for four iterations while Temporal Self-Rewarding runs for two iterations. This configuration ensures that both methods involve a comparable number of model training steps and consuming the same computation resources, allowing for a meaningful and equitable comparison. Under this setting, we demonstrate that Temporal Self-Rewarding achieves substantially better performance than the standard Self-Rewarding approach, clearly validating the effectiveness of our method despite the increased per-iteration complexity (but the same overall complexity).
>
> We greatly appreciate your thorough feedback and hope these responses address your concerns.

---

> > ### Author Rebuttal · Reviewer_cm93 · 2026-04-01
> >
> > Thanks for the rebuttal.

---

> > > ### Author Response · Authors · 2026-04-04
> > >
> > > We sincerely thank the reviewer for the thorough assessment and constructive suggestions. Your insightful feedback is invaluable in strengthening our work. We will carefully incorporate all suggestions into the final version to further improve the quality of our paper.

---

### Official Review · Reviewer_mGP9 · 2026-03-13

**Soundness:** 4
**Presentation:** 2
**Significance:** 3
**Originality:** 3
**Overall Recommendation:** 5
**Confidence:** 4

**Summary:**

The authors identify a problem in self-rewarding language models where rejected and chosen responses become similar over training iteration causing the DPO gradient to vanish. They propose the approach 'temporal self-rewarding' as a solution where they anchor rejected responses to the original SFT model (preserving low-quality negative samples) and generating chosen responses through a model trained one step ahead of current model using DPO (with chosen responses for this step generated using the current model) to produce good positive samples, the current model is then trained on this set of rejected and chosen responses to produce a final model for the next step. By doing this they maintain a quality gap between rejected and chosen examples throughout training. Their experiments across various models and benchmarks show substantial improvements over standard self-rewarding approach.

**Compliance With Llm Reviewing Policy:**

Affirmed.

**Final Justification:**

The rebuttal fully addressed my two main concerns, clarifying the notion of training collapse with additional per iteration results across model families, and providing the missing Temporal SR w/o Past ablation that confirms both components contribute meaningfully. The paper's originality and significance remain strong, with an elegant solution to a well-motivated problem and robust results across multiple models and benchmarks. While clarity could benefit from a polish to fix the noted typos, these are presentation issues that do not undermine the contribution.

**Key Questions For Authors:**

- "our theoretical analysis reveals a critical limitation: when the representational similarity between chosen and rejected responses increases, the DPO gradient vanishes, causing the training process to collapse. This theoretical prediction is empirically validated by our findings - as quantified in Fig. 1, the representations of chosen and rejected responses in the self-rewarding paradigm become progressively similar, with the score gap between them shrinking by 9 times during the same period" - Please correct me if I am wrong - the figure 1 does not seem to show training collapse. It would be helpful to maybe show an example of DPO collapse more clearly.
- Could you please reflect on why Table 2 does not include Temporal SR w/o Past?

**Limitations:**

yes

**Strengths And Weaknesses:**

### Strengths
- The problem is well-motivated and relevant and the solution proposed is simple, elegant and intuitive.
- They conduct thorough empirical evaluations - testing across various models (Qwen2.5-7B, Llama3.2-3B, Llama3.1-8B, Llama3.1-70B and Mistral-7B) of different model families and sizes. In addition, they test their models on 3 different benchmarks. Furthermore, they conduct thorough ablations showing contributions of the past and future components in their approach.
- The computational fairness argument and the judge model ablation is conducted responsibly (acknowledging the doubled per-iteration cost, and that self-rewarding may lack advantage if a stronger reward model is available).
- The results are consistent across benchmarks and show that the approach is robust.

### Weaknesses
- There are a lot of errors in the paper -
	- The following cited paper is clearly unrelated to the submitted work - Diamantidis, A. D. and Chatzoglou, P. D. Employee posttraining behaviour and performance: evaluating the results of the training process. International Journal of Training and Development, 18(3):149–170, 2014.
	- There are various typos across the paper -
		- "cosin similarity" (Figure 1 caption)
		- "evaluat them" (Section 3.7)
		- "the details of our proof is" (Section 2)
		- "no strange to understand" (Section 3.5.1)
		- DVPO is called DivPO from diverse preference optimization paper.

---

> ### Author Rebuttal · Authors · 2026-03-28
>
> We greatly appreciate your constructive suggestions and respond to your questions below.
>
> **Reviewer’s Question 1**
> "Our theoretical analysis reveals a critical limitation: ..." — Please correct me if I am wrong — the figure 1 does not seem to show training collapse. It would be helpful to maybe show an example of DPO collapse more clearly.
>
> **Our Response 1**
>
> We thank the reviewer for this important observation. By training collapse, we do not refer to it as a numerical training failure, but as a progressive loss of distinguishability between preference pairs, causing iterative training gains to saturate (we will clarify this in the revision).
>
> Figure 1 shows that under Self-Rewarding, the reward gap between chosen and rejected responses shrinks across iterations, making preference pairs less distinguishable. Consequently, while Iter0 improves a lot over SFT, later iterations show much smaller gains—a pattern we term training collapse in this context.
>
> To further support this observation, we include full experimental results for Mistral-7B and Qwen2.5-7B in addition to the Llama-3.1-8B results presented in the main experiment. The complete results are shown below. Across all model families, we observe a clear trend: after the first iteration, performance gains plateau under the Self-Rewarding paradigm, confirming that preference collapse causes the iterative training process to become ineffective.
>
>
> | Model | Method | Iter | LC Winrate | Win Rate | ArenaHard | MT-Bench |
> |-------|--------|------|------------|----------|----------|---------|
> | **Qwen2.5-7B** | sft | - | 11.45 | 7.70 | 12.70 | 5.51 |
> | | Self-Rewarding | iter0 | 19.82 | 12.92 | 18.40 | 5.93 |
> | | | iter1 | 21.66 | 15.53 | 19.50 | 6.12 |
> | | | iter2 | 20.24 | 15.65 | 22.00 | 6.00 |
> | | | iter3 | 21.53 | 18.14 | 21.50 | 6.09 |
> | | Temporal Self-Rewarding | iter0 | 27.85 | 24.78 | 27.20 | 6.25 |
> | | | iter1 | 34.01 | 35.90 | 34.40 | 6.29 |
> | **Mistral-7B** | sft | - | 12.72 | 8.45 | 6.30 | 5.28 |
> | | Self-Rewarding | iter0 | 17.98 | 15.09 | 9.10 | 5.55 |
> | | | iter1 | 18.97 | 17.70 | 9.80 | 5.35 |
> | | | iter2 | 26.15 | 24.10 | 9.80 | 5.49 |
> | | | iter3 | 25.48 | 27.58 | 12.80 | 5.68 |
> | | Temporal Self-Rewarding | iter0 | 30.11 | 32.05 | 14.00 | 5.76 |
> | | | iter1 | 30.58 | 35.16 | 15.70 | 5.49 |
> | **Llama-3.1-8B** | sft | - | 8.73 | 5.96 | 6.30 | 4.81 |
> | | Self-Rewarding | iter0 | 13.29 | 10.99 | 6.70 | 5.33 |
> | | | iter1 | 17.00 | 15.71 | 7.70 | 5.60 |
> | | | iter2 | 17.54 | 17.08 | 9.40 | 5.74 |
> | | | iter3 | 19.92 | 19.69 | 8.80 | 5.66 |
> | | Temporal Self-Rewarding | iter0 | 20.48 | 19.07 | 11.30 | 5.79 |
> | | | iter1 | 27.94 | 29.44 | 14.60 | 5.89 |
>
> **Reviewer’s Question 2**
> Could you please reflect on why Table 2 does not include Temporal SR w/o Past?
>
> **Our Response 2**
>
> Thank you for raising this important point. We agree that ablating the past component is a very valuable experiment, so we conduct additional experiments for the Temporal SR w/o Past setting, which retains the future model component while removing the past component. The updated ablation results are presented below, alongside the existing results for completeness.
>
> | Method | Iter | LC Win | Win | ArenaHard Score | MT-Bench Average |
> |--------|------|--------|-----|-----------------|------------------|
> | SFT | - | 8.73 | 5.96 | 6.3 | 4.81 |
> | **Temporal SR w/o Future&Past (Self-Rewarding)** | 0 | 13.29 | 10.99 | 6.7 | 5.33 |
> | | 1 | 17.00 | 15.71 | 7.7 | 5.60 |
> | | 2 | 17.54 | 17.08 | **9.4** | **5.74** |
> | | 3 | **19.92** | **19.69** | 8.8 | 5.66 |
> | **Temporal SR w/o Future** | 0 | 14.35 | 11.61 | 8.1 | 5.39 |
> | | 1 | 20.96 | 19.69 | 10.2 | 5.76 |
> | | 2 | 24.75 | 27.20 | 11.4 | 5.86 |
> | | 3 | **25.73** | **29.06** | **13.4** | **5.88** |
> | **Temporal SR w/o Past** | 0 | 15.21 | 12.34 | 7.2 | 5.41 |
> | | 1 | **22.47** | **23.12** | **11.8** | **5.81** |
> | **Temporal SR** | 0 | 24.08 | 19.07 | 11.3 | 5.79 |
> | | 1 | **27.94** | **29.44** | **14.6** | **5.89** |
>
> Note: For Temporal SR w/o Past, we run only two iterations like Temporal SR to ensure a fair comparison with other baselines, as the future component is similarly introduced.
>
> From the results, we observe that the past component contributes more significantly to performance gains than the future component. Nevertheless, both components individually outperform the standard Self-Rewarding baseline, and combining them in the full Temporal SR yields the best overall performance, achieving a new SOTA result.
>
> **Additional Corrections**
>
> We also appreciate the reviewer pointing out the citation error. This was indeed an oversight on our part; we mistakenly cited an irrelevant paper while reviewing literature post-training work. We have since corrected this and will ensure the revised manuscript reflects the appropriate references. In addition, we have carefully reviewed the manuscript and will correct the minor writing errors mentioned.
>
> We hope these clarifications resolve your questions.

---

> > ### Author Rebuttal · Reviewer_mGP9 · 2026-04-03
> >
> > Thank you for the comprehensive rebuttal. The additional ablation results and clarifications fully address my concerns. I will raise my score accordingly.

---

> > > ### Author Response · Authors · 2026-04-04
> > >
> > > We sincerely thank the reviewer for the positive reassessment and for raising the score. Your constructive feedback is invaluable in strengthening our work. We will carefully incorporate all suggestions into the final version to further improve the quality of our paper.

---

### Official Review · Reviewer_dHuH · 2026-03-13

**Soundness:** 3
**Presentation:** 3
**Significance:** 3
**Originality:** 3
**Overall Recommendation:** 4
**Confidence:** 4

**Summary:**

This paper identifies and analyzes the diminishing preference signal problem in iterative Self-Rewarding Language Model training and proposes the Temporal Self-Rewarding (TSR) framework to solve it. The core observation is that chosen and rejected responses progressively converge in representation similarity and score gap, causing the directional guidance term in the DPO gradient to vanish. TSR decouples the sources of chosen and rejected responses through two phases: Anchored Rejection phase anchors negative samples to the initial SFT model's outputs, while Future-Guided Chosen improves positive sample quality using a temporary future model. Experiments across three model families (Llama, Qwen, Mistral) and multiple scales validate the method's effectiveness, with significant improvements on AlpacaEval 2.0, Arena-Hard, and MT-Bench.

**Compliance With Llm Reviewing Policy:**

Affirmed.

**Final Justification:**

The rebuttal addresses most of my concerns. Thus, the reviewer tend to maintain the score of weak accept.

**Key Questions For Authors:**

1. In Phase 1, using the continuously updated $M_i$ to score $M_0$'s outputs may cause the scoring criteria for negative samples to drift across iterations. It might be interesting to see how the score of the same sample change during the iterative training. If $M_i$'s ability to identify low-quality responses changes over iterations, how is the stability of the anchoring mechanism guaranteed?

**Limitations:**

Yes.

**Strengths And Weaknesses:**

## Strengths

1. This paper provides a clear theoretical characterization of signal decay in self-rewarding training, which is new to the community. Through formal analysis of the DPO gradient, the authors demonstrate that representational convergence between chosen and rejected responses directly causes the directional term to vanish, leading to training collapse.
2. The ablation study (Table 2) effectively disentangles the contributions of the Past and Future components, demonstrating that Anchored Rejection (Past) is the primary driver of performance gains. At the same time, Future-Guided Chosen provides additional marginal benefits. This decomposition is consistent with the paper's theoretical analysis: anchoring negative samples has a more pronounced effect because it directly widens the chosen-rejected score gap, while positive sample enhancement represents incremental optimization.

## Weakness

The paper's core loop contains a key self-consistency problem: in Phase 1, the current model $M_i$ scores all responses including negative samples from $M_0$, effectively using a continuously updated model as a fixed judge. As $M_i$'s capability improves, its scoring criteria for $M_0$-generated content also shift, meaning the "anchored negative samples" are not truly anchored at the semantic level, and their score reliability drifts across iterations. The paper neither analyzes nor discusses the potential impact of this scoring inconsistency on training stability.

---

> ### Author Rebuttal · Authors · 2026-03-28
>
> Thank you for your thoughtful and constructive feedback. We greatly appreciate your recognition of our work’s strengths, particularly the theoretical characterization of the diminishing preference signal problem and the clarity of our ablation study. We hope the following explanation and additional analysis help address your concern.
>
> **Reviewer’s Weakness 1:**
> The paper's core loop contains a key self-consistency problem: in Phase 1, the current model Mi scores all responses including negative samples from M0, effectively using a continuously updated model as a fixed judge. As Mi's capability improves, its scoring criteria for M0-generated content also shift, meaning the "anchored negative samples" are not truly anchored at the semantic level, and their score reliability drifts across iterations. The paper neither analyzes nor discusses the potential impact of this scoring inconsistency on training stability.
>
> **Our Response 1:**
> Thanks again for raising this insightful question. We will answer your question from two aspects.
>
> Firstly, we evaluate the LLM-as-a-Judge performance using metrics that measure alignment with held-out human preference data, following the protocol as in the Table 4 in Self-Rewarding[1]. The results are as follows:
>
> | Metrics | SFT | SR-Iter0 | SR-Iter1 | SR-Iter2 | SR-Iter3 | TSR-Iter0 | TSR-Iter1 |
> |---------|-----|---------|---------|---------|---------|-----------|-----------|
> | Spearman Correlation | 0.632 | 0.686 | 0.708 | 0.703 | 0.712 | 0.710 | 0.722 |
> | Kendall's Tau Correlation | 0.585 | 0.621 | 0.643 | 0.638 | 0.647 | 0.637 | 0.653 |
> | Pairwise Accuracy | 0.662 | 0.667 | 0.663 | 0.670 | 0.672 | 0.667 | 0.670 |
> | Max Score Accuracy | 0.711 | 0.756 | 0.803 | 0.815 | 0.821 | 0.811 | 0.832 |
> | Exact Match | 0.198 | 0.204 | 0.207 | 0.213 | 0.210 | 0.210 | 0.216 |
>
> As the table shows, the judge capability improves during iteration which means a positive correlation between generation and judgment capabilities. This aligns with the core finding of Self-Rewarding that stronger generative capability during the iteration means stronger judging ability, motivating the use of the evolving model as a judge rather than a fixed one in the Self-Rewarding structure.
>
> Secondly, to demonstrate the effectiveness of our framework does not rely on the evolving judge, we have conducted an ablation study in our paper where we fix the judge model as shown in Figure 2. The results show that Temporal Self-Rewarding remains stable and significantly outperforms the Self-Rewarding baseline with a fixed judge. This confirms that our mechanism is robust and does not depend on the iterative update of the judge model for stability.
>
> As the reviewer's question mirrors the stated weakness, please refer to our response above.
>
>
> We hope this analysis clarifies the stability of our anchoring mechanism and addresses your concerns.
>
> ---
>
> **References**
>
> [1] Yuan, Weizhe, Richard Yuanzhe Pang, Kyunghyun Cho, Xian Li, Sainbayar Sukhbaatar, Jing Xu, and Jason E. Weston. "Self-rewarding language models." In *Forty-first International Conference on Machine Learning*, 2024.

---

### Decision · Program_Chairs · 2026-04-30

**Decision:**

Accept (regular)

**Comment:**

The paper studies a flaw in self-rewarding models related to mode collapse and a reduced learning signal. They introduce a novel method to maintain diversity in the quality of preferences generated by the model and demonstrate that this leads to performance improvements.

Reviewers appreciated the theoretical analysis of the paper and the strength of the results. Reviewers raised concerns about the comprehensiveness of the evaluation, the presentation of the work, and some properties of the method. The authors addressed these in the rebuttal --- reviewers uniformly accepted the responses and adjusted scores accordingly.

This is a paper with agreement from the reviewers on the underlying quality and significance of the results. Based on that, I recommend acceptance of the paper.